# Analyzing the Impact of Diesel Exhaust Particles on Lung Fibrosis Using Dual PCR Array and Proteomics: YWHAZ Signaling

**DOI:** 10.3390/toxics11100859

**Published:** 2023-10-13

**Authors:** Byeong-Gon Kim, Pureun-Haneul Lee, Jisu Hong, An-Soo Jang

**Affiliations:** Division of Allergy and Respiratory Medicine, Department of Internal Medicine, Soonchunhyang University Bucheon Hospital, 170 Jomaru-ro, Wonmi-gu, Bucheon 14584, Republic of Korea; byeonggone@naver.com (B.-G.K.);

**Keywords:** diesel exhaust particulate matter, fibrosis, PCR array, proteomics, miRNA

## Abstract

Air pollutants are associated with exacerbations of asthma, chronic bronchitis, and airway inflammation. Diesel exhaust particles (DEPs) can induce and worsen lung diseases. However, there are insufficient data to guide polymerase chain reaction (PCR) array proteomics studies regarding the impacts of DEPs on respiratory diseases. This study was performed to identify genes and proteins expressed in normal human bronchial epithelial (NHBE) cells. MicroRNAs (miRNAs) and proteins expressed in NHBE cells exposed to DEPs at 1 μg/cm^2^ for 8 h and 24 h were identified using PCR array analysis and 2D PAGE/LC-MS/MS, respectively. *YWHAZ* gene expression was estimated using PCR, immunoblotting, and immunohistochemical analyses. Genes discovered through an overlap analysis were validated in DEP-exposed mice. Proteomics approaches showed that exposing NHBE cells to DEPs led to changes in 32 protein spots. A transcriptomics PCR array analysis showed that 6 of 84 miRNAs were downregulated in the DEP exposure groups compared to controls. The mRNA and protein expression levels of *YWHAZ*, *β-catenin*, *vimentin*, and *TGF-β* were increased in DEP-treated NHBE cells and DEP-exposed mice. Lung fibrosis was increased in mice exposed to DEPs. Our combined PCR array–omics analysis demonstrated that DEPs can induce airway inflammation and lead to lung fibrosis through changes in the expression levels of *YWHAZ*, *β-catenin*, *vimentin*, and *TGF-β*. These findings suggest that dual approaches can help to identify biomarkers and therapeutic targets involved in pollutant-related respiratory diseases.

## 1. Introduction

According to the World Health Organization (WHO), respiratory diseases caused by air pollution are increasing worldwide [1]. Strong associations are evident between air pollution and lung dysfunction, airway inflammation, the worsening of asthmatic symptoms, and the increased numbers of hospital outpatient visits by those with respiratory diseases [2]. Worldwide, air pollution causes many deaths annually [2]. Many studies have revealed associations between respiratory diseases and air pollution. The prognoses of such diseases are poor, and their treatment and management are costly [3]. The sources of air pollution vary from local cigarette and coal smoke and industrial dust to widespread emissions from engines and industrial sources. Air pollutants include particulate matter ≤2.5 or ≤10 µm in diameter (PM 2.5 and 10, respectively) as well as finer diesel exhaust particles (DEPs), factory pollutants, nitrogen dioxide, ozone, and aeroallergens [4]. Air pollutants penetrate deeply into airways, reaching the bronchioles and alveoli and then entering the bloodstream to trigger organ inflammation, dysfunction, and toxicity. Pollutants accumulating in the lungs exacerbate the symptoms of respiratory disease and may kill elderly subjects and young children [5]. DEPs and industrial dust increase the global disease burden [6,7]. DEPs include soot, aerosols, zinc, magnesium, oils, cerium, iron, carbon, beryllium, manganese, platinum, copper, acetaldehyde, acrolein, cobalt, beryllium, dioxins, nitrogen oxides, benzene, cyanide, biphenyls, chlorine, phenol, selenium, and volatile organic compounds [6,7,8,9,10], which affect airway epithelial cells and regulatory T cells within the innate immune system, resulting in adaptive immunity activation in airways. Animal models have revealed that DEPs play important roles in the development of chronic lung injuries, inducing airway hyperresponsiveness (AHR), inflammation, asthma, and the exacerbation of chronic obstructive pulmonary disease (COPD) [7,11,12].

Proteomics and polymerase chain reaction (PCR)-based methods have facilitated modern work on respiratory diseases. Proteomics methods allow the identification of changes in protein expression levels in disease and their utilization as biomarkers of disease [13,14].

MicroRNAs (miRNAs) are single-stranded noncoding RNAs approximately 22 nucleotides in length that regulate gene transcription by binding to the 3′ untranslated regions of mRNAs [15,16]; miRNAs play important roles in disease by modulating signal transduction pathways [17,18]. The various targets and functions of miRNAs are under active exploration [19,20]. miRNA expression is tissue-specific [21], and it is affected by irritants and other materials [22]. However, single omics analyses of miRNA–mRNA and protein–protein interactions reflecting the harmful effects of air pollutants on the respiratory tract have been insufficient to identify disease biomarkers [23,24].

Therefore, the aim of this study was to identify genes and proteins in bronchial epithelial cells and mice exposed to DEPs using a PCR array–proteomics analysis.

## 2. Materials and Methods

### 2.1. Preparation of Diesel Exhaust Particulate Matter

DEPs (SRM 2975, Industrial Forklift) were purchased from the National Institute of Standards and Technology (Gaithersburg, MD, USA) for experimental mouse lung tissue and human bronchial cell stimulation. The DEPs were sterilized using an autoclave and suspended in serum-free media after coating with BSA to minimize particle aggregation and hydrophobicity [25].

### 2.2. NHBE Cell Culture and Stimulation with DEPs

Normal human bronchial primary epithelial cells (NHBE) (3500 cells/cm^2^, CC-2540, Lonza, Basel, Switzerland) were grown in a BEGM BulletKit (Lonza, Basel, Switzerland). The medium was replaced every 48 h until the cells reached 90% confluence at 37 °C in 5% CO_2_. The cells were seeded in 6-well plates. Then, 24 h prior to the experiment, they were stimulated with DEPs at a concentration of 1 μg/cm^2^ with the proteomics or microRNA samples for 8 or 24 h.

### 2.3. NHBE Cell Viability Assays

NHBE cell viability was measured using cell counting kit-8 (CCK-8) (Dojindo, Tokyo, Japan) according to the manufacturer’s instructions. Briefly, 10 μL of the CCK-8 solution was added to each well and incubated at 37 °C in an environment with 5% CO_2_ for 2 h. The absorbance at 450 nm was measured with a microplate reader. The experiments were performed at least three times.

### 2.4. Two-Dimensional (2D) Electrophoresis and Image Analysis in NHBE Cells

NHBE cells were harvested via centrifugation and then disrupted with a lysis buffer containing 5 mM Tris-HCl (pH 7.4), 100 mM NaCl, 1% Triton X-100, and 2 mM PMSF. The cell lysate was centrifuged at 12,000× *g* for 30 min, and the supernatant fraction was collected. The protein concentration was determined using a commercial BCA assay kit (Thermo, Chicago, IL, USA), and the samples were stored at −70 °C until use. Immobiline DryStrips (Amersham Biosciences, Piscataway, NJ, USA) were used for isoelectric focusing, which was carried out with 1 mg of the protein on an IPGphor system (Amersham Biosciences). After IEF separation, they were separated in the second dimension using SDS-PAGE. For the image analysis, they were visualized using the Coomassie Brilliant blue G-250 staining method. Next, the 2D spot intensity was calculated by integrating the optical density over the spot area. The values were normalized and then exported to the statistical analysis software. The experiments were performed at least two times [26].

### 2.5. Protein Identification via LC-MS/MS Analysis

Peptides obtained after trypsin digestion from NHBE cell lysates were analyzed with a Nano LC-MS/MS. The Nano LC-MS/MS analysis was performed with an Easy n-LC (Thermo Fisher, Carlsbad, CA, USA) and an LTQ Orbitrap XL mass spectrometer (Thermo Fisher, Carlsbad, CA, USA) equipped with a nano-electrospray source. Digested samples were dissolved in 0.1% TFA, and 1 μL (equivalent to 2 μg) was injected into a C18 nanobore column (150 mm × 0.1 mm, 3 μm pore size; Agilent). Mobile phase A for the LC separation was 0.1% formic acid and 3% acetonitrile in deionized water, and mobile phase B was 0.1% formic acid in acetonitrile. The chromatography gradient was designed for a linear increase from 0% B to 60% B in 6 min, from 60% B to 90% B in 1 min, and to 3% B in 8 min. The flow rate was maintained at 1800 nL/min. Mass spectra were acquired using data-dependent acquisition with a full mass scan (380–1700 m/z) followed by 10 MS/MS scans. For MS1 full scans, the orbitrap resolution was 15,000 and the AGC was 2 × 10^5^. For MS/MS in the LTQ, the AGC was 1 × 10^4^ [27].

### 2.6. Database Search and Protein Identification

The mascot algorithm (Matrix science, MA, USA) was used to identify MS and MS/MS peptide sequence data present in a protein sequence database. Additional peptide sequence data were used in subsequent searches. The Mascot databases (Matrix Science; http://www.matrixscience.com (accessed on 21 April 2017)), Swiss-prot database, and NCBI database were searched using the MSDB protein sequence database for human protein. The database search criteria were taxonomy, homo sapiens; fixed modification, carbamidomethylated at cysteine residues; variable modification, oxidized at methionine residues; maximum allowed missed cleavage, 2; MS tolerance, 10 ppm; and MS/MS tolerance, 0.8 Da. The peptides were filtered with a significance threshold of *p* < 0.05. And adjusted proteins were visualized in various colors [28].

### 2.7. miRNA Extraction and cDNA Synthesis

A 700 μL volume of QIAzol Lysis Reagent (Qiazen, Chatsworth, CA, USA) was added to the sample of NHBE cells and incubated at room temperature for 5 min. The microRNA was extracted from the NHBE cells with an miRNeasy Mini Kit 50 (Qiazen), according to the manufacturer’s instructions. Extracted microRNA was eluted in 14 μL of RNase-free water. The concentration of the microRNA was normalized by measuring the absorbance at 260 nm in a NanoDrop spectrophotometer. The extracted miRNA was reverse-transcribed into cDNA using an miScript II RT Kit (Qiagen). The experiments were conducted according to the manufacturer’s instructions. Briefly, each microRNA sample was used for cDNA synthesis with 5xHiFlex Buffer in reverse-transcription reaction master mixes. For the cDNA synthesis reaction, the PCR settings were 60 min at 42 °C and heat inactivation of reverse transcriptase for 5 min at 95 °C. Each cDNA sample was further diluted according to the PCR array protocol and stored at −20 °C until use.

### 2.8. Total RNA Extraction and Polymerase Chain Reaction (PCR)

Total RNA was extracted using a QIAzol lysis reagent (Qiagen) according to the manufacturer’s instructions. RNA was reverse-transcribed via incubation with 0.5 mM dNTP, 2.5 mM MgCl_2_, 5 mM DTT, 1 µL of oligo DT (0.5 μg/µL), and SuperScript II RT (Invitrogen-Life Technologies Corp, Carlsbad, CA, USA) at 42 °C for 50 min and heat-inactivated at 70 °C for 15 min. PCR was performed using AccuPower PCR Premix (Bioneer, Daejeon, Republic of Korea). The PCR products were analyzed via electrophoresis using a 2.0% agarose gel. The levels of expression of mRNA were normalized to the housekeeping gene *β-actin*. The sequences of the primers of *YWHAZ*, *β-catenin*, *vimentin*, *transforming growth factor* (*TGF*)-*β*, and *β-actin* are listed in Table 1.

### 2.9. Profiling of PCR Array

The synthesized cDNA was mixed with the RT^2^ SYBR Green Master mix (Qiagen). The experiment was performed according to the manufacturer’s instructions. The PCR array contained 84 miRNAs that were involved with 12 constitutively active genes as an internal control. The PCR array was performed on an Applied StepOnePlus Real-Time PCR System (Applied Biosystems, Foster City, CA, USA) according to the RT^2^ Profiler PCR Array instructions under the following conditions: 95 °C for 15 min, then 40 cycles at 94 °C for 15 s, 55 °C for 30 s, and 70 °C for 30 s. The PCR array data analysis and calculations were conducted using a PCR array data analysis tool provided by the manufacturer. The threshold cycle (CT) values were normalized based on an automatic selection from a housekeeping gene (HKG) panel of reference genes using the QIAGEN website’s Data Analysis Center (https://www.qiagen.com/us/shop/genes-and-pathways/data-analysis-center-overview-page/ (accessed on 18 August 2017)). Differences in expressed miRNAs were selected between the DEP exposure sample and the control sample. The experiments were performed at least two times.

### 2.10. miRNA Target Prediction and Bioinformatics Data Analysis

Target genes of miRNAs were predicted using an online target prediction database. The analysis of miRNA gene prediction used miRTarBase (www.mirtarbase.mbc.nctu.edu.tw (accessed on 21 August 2017)) and miRDB (http://www.mirdb.org (accessed on 21 August 2017)). The overlapping gene in the miRNA target gene databases was selected for further analysis. A gene ontology (GO) and Kyoto Encyclopedia of Genes and Genomes (KEGG) analysis was performed using an online database of molecular functions, biological processes, cellular components, and pathway enrichment [29,30]. The Database for Annotation, Visualization, and Integrated Discovery (DAVID) [31] bioinformatics resources were used for the GO annotation and KEGG pathway analysis. *p* value < 0.05 was set as the threshold. The miRNA–miRNA or miRNA target prediction data network was then edited, and the interactions were visualized using Cytoscape (version: 3.5.1) software [30].

### 2.11. Design of In Vivo Experiment

Female BALB/c mice (*n* = 30 in each group), 6 weeks of age and free of mouse-specific pathogens, were obtained from Orient Bio (Orient Bio Inc., Seongnam, Republic of Korea). The particle inhalation methods for mice were conducted as in previously studies [7,32]. The mice were housed throughout the experiments in a laminar flow cabinet and maintained on standard laboratory chow ad libitum. The mice were exposed to 100 μg/m^3^ DEPs in a closed-system chamber attached to an ultrasonic nebulizer (NE-UO7; Omron Corporation, Tokyo, Japan) with an output of 1 mL/min and a 1 to 5 µm particle size. The control mice were administered and exposed to a saline solution alone. The mice were exposed to DEPs for 1 h a day for 5 days a week from 4 and 8 weeks, and on days 27 and 55 AHR was measured. On days 28 and 56, BALF was collected and tissue was processed for protein immunoblotting. After ligation of the right main bronchus, the left lung was fixed with 4% paraformaldehyde in phosphate-buffered saline and paraffin-embedded for immunohistochemistry. All experimental animals in this study were used according to the guidelines of the Institutional Animal Care and Use Committee (IACUC) at Soonchunhyang University Medical School. 

### 2.12. Determination of Airway Responsiveness to Methacholine

AHR was measured in unrestrained, conscious mice 1 day after the last challenge, as previously described [7,32,33]. Mice were placed in a barometric plethysmographic chamber (All Medicus Co., Ltd., Seoul, Republic of Korea), and baseline readings were taken and averaged for 3 min. Aerosolized methacholine (Sigma-Aldrich, St. Louis, MO, USA) in increasing concentrations (from 2.5 to 50 mg/mL) was nebulized through an inlet of the main chamber for 3 min. Readings were taken and averaged for 3 min after each nebulization, at which time the enhanced pause (Penh) was determined. The Penh, calculated as (expiratory time/relaxation time − 1) × (peak expiratory flow/peak inspiratory flow) according to the manufacturers’ protocol, is a dimensionless value that represents the proportion of the maximal expiratory to maximal inspiratory box pressure signals and the timing of expiration. The Penh is used as a measure of airway responsiveness to methacholine. The results are expressed as the percentage increase in Penh following a challenge with each concentration of methacholine, where the baseline Penh (after saline challenge) is expressed as 100%. Penh values averaged for 3 min after each nebulization were evaluated.

### 2.13. Bronchoalveolar Lavage Fluid Morphology Analysis

Mice were anesthetized the next day, and BALF was obtained and stored at −20 °C until use. Morphology analyses in inflammatory cells from the BALF were conducted as in previously studies [7,32,33]. BALF differential cell counts were performed using Diff-Quick-staining cytospin slides, with 500 cells counted per animal. A portion of the lung was fixed in 4% buffered paraformaldehyde and embedded in paraffin. The tissue was cut into 4 μm slices and used for the histological analysis.

### 2.14. Preparation of Lung Tissues for Histology

Trachea and lung tissues were removed from the mice. A 4% paraformaldehyde fixing solution was infused into the lungs via the trachea. The specimens were dehydrated and embedded in paraffin. For the histological examination 4 µm sections of embedded tissue were cut on a rotary microtome, placed on glass slides, deparaffinized, and stained sequentially with hematoxylin and eosin [7].

### 2.15. Western Blotting

Western blot and protein extraction methods were used on the lung tissue as in previously studies [7,34]. The NHBE cells and mouse lung tissues were collected and stored at −80 °C until use. Briefly, NHBE cells and mouse lung tissues were collected with an MEM kit (Thermo, Chicago, IL, USA) and an M-PER kit (Thermo) with protease and phosphatase inhibitors (Roche, IN, USA). Protein levels were quantified using a Pierce BCA Protein Assay Kit (Thermo). Proteins were separated via SDS-PAGE and transferred to PVDF membranes (Millipore Corp., Boston, MA, USA). The membranes were blocked with 5% BSA (GenDEPOT, TX, USA) or 5% skim milk (BD Difco, Franklin Lakes, NJ, USA) in 0.1% Tween 20 in TBS for 2 h at room temperature. And the membranes were incubated with rabbit anti-*YWHAZ* (1:500, Abcam, Boston, MA, USA), mouse anti-*β-catenin* (1:500, Santa Cruz Biotechnology, Santa Cruz, CA, USA), mouse anti-*vimentin* (1:500, Santa Cruz Biotechnology), rabbit anti-*TGF-β* (1:1000, Abcam), and mouse anti-*β-Actin* (1:5000, Sigma-Aldrich) antibodies overnight at 4 °C. The next day, the membranes were incubated with HRP-conjugated secondary antibodies. Detection was performed using an EzWestLumi plus Western blot detection reagent (ATTO Corporation, Tokyo, Japan). The relative abundance of protein was determined using quantitative densitometry data that were normalized to *β-actin* (Sigma-Aldrich).

### 2.16. Immunohistochemistry

Lung tissue was stained as previously described [7]. Briefly, lung tissue slides were deparaffinized and rehydrated in an ethanol series. Next, they were treated with 1.4% H_2_O_2_ (hydrogen peroxide) in methanol for 30 min to block endogenous peroxidase, and then non-specific binding was blocked using normal mouse serum. Then, the slides were incubated with the rabbit anti-*YWHAZ* antibody (1:50, Abcam). The next day, the sections were incubated with an ABC kit (Vector Laboratories, Newark, CA, USA). The color reaction was developed by staining with a liquid DAB+ substrate kit (Golden Bridge International Inc, Mukilteo, WA, USA). After immunohistochemical staining, the slides were counterstained with Herris’s hematoxylin for 1 min. Prior to observation, the samples were mounted with a cover slip using mounting media (Amresco, Solon, OH, USA). The immunohistochemical expression was quantified using ImageJ 1.51b (National Institutes of Health, Bethesda, MD, USA).

### 2.17. Masson’s Trichrome Statin

The lung tissue slides were deparaffinized and rehydrated in an ethanol series. Then, the slides were stained in accordance with the manual of the Masson trichrome assay kit (Sigma-Aldrich). Briefly, Bouin’s solution was used as a mordant to intensify the color reactions. The nucleus and cytoplasm were dyed using hematoxylin and scarlet acid. A phosphotungstic/phosphomolybdic acid solution was used to change a positive charge to a negative charge. After staining with aniline blue, the samples were treated with 1% acetic acid [7]. The positive trichrome-stained area was quantified using ImageJ software (National Institutes of Health). Values are reported as the percentage of positive areas within the total sample.

### 2.18. Statistical Analysis

The statistical analysis was performed using the SPSS statistical software package (ver. 20.0; SPSS; Chicago, IL, USA). All data are expressed as means ± standard deviations (SDs) or SEMs. Group differences were compared using a two-sample t test, Mann–Whitney test, or Pearson’s χ^2^ test for normally distributed, skewed, and categorical data, respectively. A one-way ANOVA or a two-way ANOVA was used for methacholine data and multiple DEP exposure data. Values of *p* < 0.05 were deemed to indicate statistical significance [7].

## 3. Results

### 3.1. NHBE Cell Viability

Figure 1 shows a flowchart of our experimental approach. Normal human bronchial epithelial (NHBE) cells were exposed to DEPs at concentrations of 0.2, 1, 2, 5, and 10 µg/cm^2^, and cell viability was estimated using a CCK-8 assay. The viabilities of cells exposed to DEPs at 1 µg/cm^2^ for 8 h and 0.2 µg/cm^2^ for 24 h were not significantly different (Figure 1); however, exposure to DEPs at 2 µg/cm^2^ for 8 h and 1 µg/cm^2^ for 24 h significantly reduced cell viability compared to control (both *p* < 0.05, Figure 1).

### 3.2. Two-Dimensional PAGE and LC-MS/MS in NHBE Cells

To explore the protein expression levels in NHBE cells after 8 and 24 h of exposure to DEPs at 1 µg/cm^2^, cytosolic fractions were obtained via differential centrifugation and subjected to two-dimensional (2D) polyacrylamide gel electrophoresis (PAGE) (six replicates/treatment). Thirty-two spots (Table 2), the intensities of which changed by 1.5–2-fold following DEP exposure, were detected on each gel (Figure 2); all proteins had pI values of 3–10 and molecular masses of 10–150 kDa. The 32 protein spots with differential expression according to DEP exposure were excised and digested with trypsin. The peptides were subjected to liquid chromatography with tandem mass spectrometry (LC-MS/MS), and the proteins were identified (Table 2 and Figure 3). The identified proteins included a multifunctional enzyme; a calcium-binding protein; a cytoskeletal protein; and proteins involved in TGF-β signaling, energy-dependent processes, glucose regulation, regulation of mitotic translation, signal transduction, actin binding, tumor necrosis, macromolecular assembly, T-lymphocyte signaling, phosphorylation, phospholipid binding, glycolysis, eukaryotic DNA replication, bisphosphoglycerate metabolism, proteolysis, cell division, and chloride channel function (Table 2).

### 3.3. miRNA Expression Levels in NHBE Cells

Figure 4 shows a scatter plot indicating the miRNA changes outlined in Table 3 (determined using the Qiagen method; https://www.qiagen.com (accessed on 18 August 2017)) in NHBE cells exposed to DEPs. Following the exposure of NHBE cells to DEPs for 8 h, 22 and 11 miRNAs were upregulated and 17 and 7 miRNAs were downregulated by 1.5- and 2.0-fold, respectively; following DEP exposure for 24 h, 26 and 7 miRNAs were upregulated and 20 and 8 were downregulated by 1.5- and 2.0-fold, respectively (Table 4). When the two sample sets were compared, the levels of six miRNAs changed significantly at both times (Figure 5). According to the particle exposure time, mRNAs predicted to share miRNAs were subjected to filtration prior to an overlap analysis with proteins. To reduce the standard deviations of factors affected by DEPs in the overlap analysis, six miRNAs with comparable changes after 8 and 24 h of exposure were analyzed.

### 3.4. Predicted Targets of miRNAs of Interest

We used the miRDB and TargetScan databases to predict the targets of hsa-miR-22-3p, hsa-miR-18-3p, hsa-miR-30c-5p, hsa-miR-106b-5p, hsa-let-7a-5p, and hsa-miR-146a-5p. These miRNAs targeted 42, 56, 206, 19, 29, and 261 genes, respectively. Both scans indicated that *YWHAZ* was a target of three miRNAs (hsa-miR-22-3p, hsa-miR-18-3p, and hsa-miR-30c-5p), in particular at seven miRNAs, which also showed increased expression of the *YWHAZ* gene in the proteomics data. A gene ontology (GO) enrichment analysis indicated that the *YWHAZ* gene is involved in apoptosis modulation and signaling as well as the regulation of Wnt-mediated *β-catenin* signaling and the transcription of target genes. The dual approach method helped to evaluate the flow of genotype and phenotype information from PCR arrays to proteomics analyses or vice versa, enabling the detection of the *YWHAZ* gene.

### 3.5. Genes Targeted by the Selected miRNAs and GO and KEGG Pathway Analyses

Genes targeted by hsa-miR-22-3p, hsa-miR-18-3p, and hsa-miR-30c-5p, including the *YWHAZ* gene, are involved in apoptosis, *TGF-β* signaling, and cancer (Figure 6 and Figure 7). The functions of genes targeted by hsa-miR22-3p, hsa-miR-30c-5p, and hsa-miR-18a-3p were explored via a KEGG pathway and GO analysis using the DAVID database. The top 20 enriched KEGG pathways and top 10 GO clusters are shown in Figure 7A. The *YWHAZ* gene was associated with both *TGF-β* signaling and cancer. The top 10 GO clusters were associated with molecular functions, biological processes, and cellular components.

### 3.6. DEPs Triggered YWHAZ, β-Catenin, Vimentin, and TGF-β mRNA and Protein Expression in NHBE Cells

To investigate the effects of DEP exposure time on *YWHAZ* expression in NHBE cells, we analyzed *YWHAZ*, *β-catenin*, *vimentin*, and *TGF-β* expression and signaling pathways among RNA and protein samples collected from cells exposed to DEPs for 8 and 24 h (Figure 8). The levels of *YWHAZ*, *β-catenin*, *vimentin*, and *TGF-β* mRNA and protein expression increased significantly following the exposure of NHBE cells to DEPs in comparison to untreated controls, as determined via PCR and Western blot analyses (all *p* < 0.05) (Figure 8).

### 3.7. Expression of YWHAZ, β-Catenin, Vimentin, and TGF-β in DEP-Exposed Mice

To examine the effects of DEP exposure on AHR (Figure 9A), airway resistance was measured in mice using invasive whole-body plethysmography. AHR increased in mice that inhaled DEPs. In comparison with the control group (Figure 9B), exposure to DEPs substantially increased the numbers of macrophages and neutrophils, leading to a mixed inflammatory response (Figure 9C). These observations indicated that DEP exposure induced airway inflammation in the mouse lung; the inflammation was more noticeable in the 8-week exposure group than in the 4-week exposure group (Figure 9B,C).

The levels of *YWHAZ*, *β-catenin*, *vimentin*, and *TGF-β* expression at both the mRNA and protein levels in the lungs of mice increased significantly following DEP treatment, as determined via PCR, Western blotting, and immunohistochemical analyses (Figure 10). The levels of all proteins were higher in DEP-treated mice than in sham-treated mice (all *p* < 0.05) (Figure 10B) but did not differ significantly between 4 and 8 weeks of DEP exposure. Immunohistochemical analyses showed that the levels of all proteins were higher in the lungs of DEP-treated mice than in sham-treated mice, with the lungs in the exposure groups exhibiting many bronchial foci of inflammatory cells (Figure 10). However, the extent of DEP toxicity did not differ significantly between the 4- and 8-week exposure groups.

### 3.8. Increased Lung Fibrosis Following DEP Inhalation

To confirm the toxic effects of DEPs on lungs, we observed lung tissue under a light microscope. No obvious fibrosis was apparent in lungs of mice that received saline at 4 and 8 weeks (Figure 10C), but fibrosis was evident in the lungs of test mice (Figure 10C). Mouse body weight was significantly lower in the DEP group than in the control group (Figure 11A). However, mouse lung weight was significantly greater in the 8-week DEP group than in the control group (Figure 11B). The lesional areas increased significantly in the latter lungs (Figure 10C), correlating with the increases in lung weight and fibrosis (Figure 11 and Figure 12). In this study, the RNA and protein levels of *TGF-β* and *vimentin* were significantly increased by DEP exposure in both NHBE cells and mice compared to controls (Figure 9 and Figure 10). Fibrosis could be quantified in DEP-exposed mouse lung tissue due to significant increases in lung weight and the extent of collagen staining in the epithelium (Figure 10C and Figure 11B).

## 4. Discussion

Air pollutants have been shown to trigger airway diseases and cancer, and to exacerbate the decline in pulmonary function with age [7,9,10]. DEPs are air pollutants consisting of soot, aerosols, zinc, magnesium, oils, cerium, iron, carbon, manganese, platinum, copper, acetaldehyde, acrolein, cobalt, beryllium, dioxins, nitrogen oxides, benzene, chlorine, and phenol [35,36]. In this study, DEPs triggered airway inflammation and airway obstruction in an animal model (Figure 9B,C), suggesting that the inhalation of air pollutants can cause lung inflammation and dysfunction. The levels of *IL-5*, *IL-6*, *IL-10*, *IL-13*, *interferon-γ*, and *TGF-β* increase after long-term DEP exposure, suggesting that these Th2 and Th1 cytokines trigger airway inflammation after DEP exposure [7,36]. Long-term inhalation of DEPs triggers lung cancer and cardiopulmonary disease [37]. DEPs also exacerbate allergic airway diseases, such as asthma, rhinitis, and COPD, by inducing neutrophilic inflammation and lung dysfunction [32,35,38,39]. NHBE cell exposure to DEPs activated several signaling pathways involved in the induction of apoptosis, fibrosis, and cell death; the relevant genes act within the cell cycle, as signal transducers, or as cytoskeletal components (Figure 3, Figure 7, and Figure 8). Comparable changes were observed in the lungs of mice following DEP inhalation (Figure 9, Figure 10 and Figure 11). There was overlap in the predicted target genes of DEP-induced miRNAs among proteomics, PCR, and Western blot analyses (Figure 8 and Figure 10). We identified 6 miRNAs and 32 proteins with differential expression in DEP-exposed cells and mice. One of the genes identified in our study, *YWHAZ*, has been shown to exacerbate disease and is associated with poor outcomes in cancer patients [40]. *YWHAZ*, a member of the 14-3-3 family, affects several signaling pathways by interacting with phosphoserine- and serine-containing proteins, including *β-catenin*, which is involved in fibrosis and cancer [41]. Although an interaction between *β-catenin* and *YWHAZ* has been reported, the mechanism by which *YWHAZ* acts following air pollutant inhalation remains unclear. *β-Catenin* activates Wnt signaling in patients with respiratory disease [42]. The *Wnt*/*β-catenin* combination triggers, exacerbates, and induces the progression of tumorigenesis, idiopathic pulmonary fibrosis, and COPD [43]. *Wnt*/*β-catenin* signaling also regulates the inflammatory response to pathogenic bacteria [44]. Such signaling either increases or decreases *NF-κB* signaling, and Toll-like receptor signaling modulates *Wnt*/*β-catenin* signaling [42,43]. Increased *vimentin* levels are correlated with the tumorigenesis, invasion, and metastasis of several types of cancer, including lung cancer [45]. *Vimentin* interacts with the 14-3-3 protein to modulate tumorigenesis, as do *YWHAZ* and *β-catenin*. *TGF-β* released during cancer progression increases the levels of *β-catenin* and *vimentin* [45,46] and promotes tumorigenesis and lung tissue fibrosis by upregulating collagen and *YWHAZ* expression.

Our complementary PCR and proteomics approaches identified signaling pathways triggering *YWHAZ*, *β-catenin*, *vimentin*, and *TGF-β* overexpression in lung tissue following exposure to DEPs. Changes in miRNA levels following DEP inhalation triggered apoptosis and *TGF-β* signaling. In addition, there was overlap in the predicted target genes between the mRNA and proteomics data, particularly in the context of *YWHAZ*. Mechanistic analyses showed that the *vimentin* and *TGF-β* levels increased due to interactions among *YWHAZ*, *β-catenin*, *vimentin*, and *TGF-β*, all of which are involved in the induction of lung fibrosis (Figure 10C) [41,42,47]. We found that the levels of *YWHAZ*, *β-catenin*, *vimentin*, and *TGF-β* expression at both the mRNA and protein levels were increased in NHBE cells and mice exposed to DEPs, suggesting that DEPs trigger airway inflammation, fibrosis, and apoptosis (Figure 8, Figure 9, Figure 10, Figure 11 and Figure 12). *Wnt*/*β-catenin*, *vimentin*, *TGF-β*, and collagen are all expressed in the lungs. In the context of pulmonary fibrosis, *YWHAZ* can regulate genes involved in various types of signal transmission. Notably, we found that DEPs induced fibrosis-related changes, along with increases in the levels of collagen, *β-catenin*, *vimentin*, and *TGF-β*; these findings suggest the involvement of endogenous Wnt/β-catenin and *YWHAZ*-induced *vimentin* in pulmonary fibrosis pathways [41,42,47]. DEPs increase airway inflammatory cell infiltration, airway resistance, and respiratory disease symptoms. Our results provided novel insights into the pathogenesis of air pollution and identified potential therapeutic targets for the treatment of patients with respiratory diseases.

The roles played by miRNAs in the context of respiratory diseases remain poorly understood. In general, few miRNAs are expressed in affected cells, and single miRNAs regulate the expression of hundreds (or even thousands) of mRNAs. Therefore, miRNA–mRNA interactions that are significant in the context of respiratory disease remain unclear [47]. We presented complementary mRNA and proteomics data. Although electrophoretic data remain valuable, PAGE has certain limitations, including the inability to detect membrane and hydrophobic proteins and proteins with high or low pI values and molecular weights [27]. Therefore, Western blot data are also required. Air pollutants have been shown to induce respiratory disease, but the mechanisms remain unclear. Complementary analyses are appropriate when seeking to predict associations between expressed miRNAs and the effects of air pollutants. Dual approaches using methods to study biological pathways can improve the accuracy of predictions regarding disease phenotypes and genotypes, thus aiding in the development of pathway biomarkers and therapeutic targets. We identified 6 miRNAs and 32 proteins with differential expression between DEP-exposed and control cells. Further data from disease outbreaks will be valuable for confirming our findings. The linking of PCR array data to proteomics data, as used in the present study, represents a novel and robust approach with the potential to aid in the development of new therapies for respiratory diseases.

## Figures and Tables

**Figure 1 toxics-11-00859-f001:**
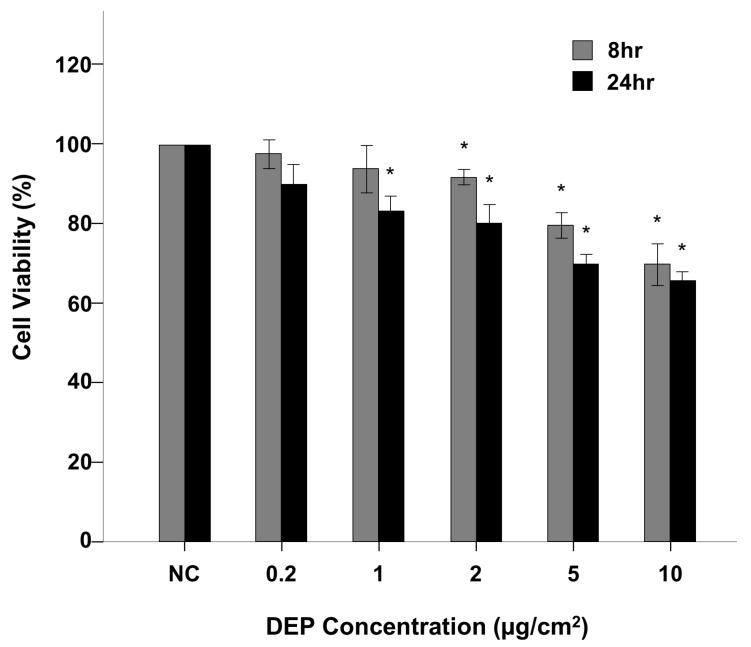
DEP-exposed NHBE cell viability assessed using CCK-8 assay. The CCK-8 assay showed that DEP exposure decreased cell viability in NHBE cells. * *p* < 0.05 vs. the control group.

**Figure 2 toxics-11-00859-f002:**
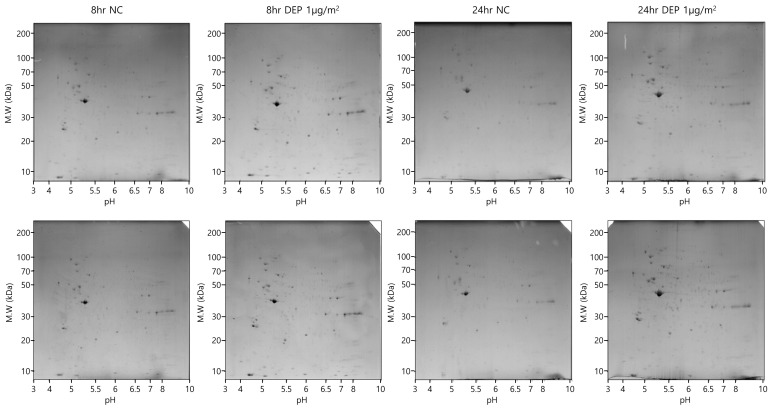
Two-dimensional electrophoresis in DEP-exposed NHBE cells. The 2D PAGE image from lysates of untreated cells was used as a master gel and reference map.

**Figure 3 toxics-11-00859-f003:**
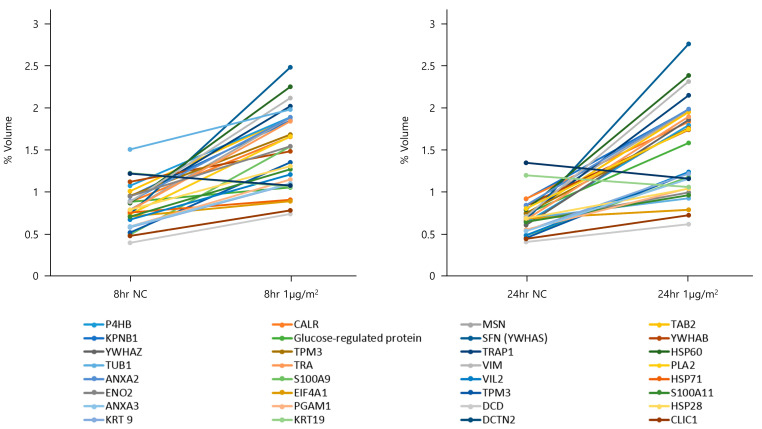
Cluster analysis of 32 proteins with significant differential expression caused by DEP treatment of NHBE cells at 1 μg/cm^2^ for 8 and 24 h. The profile of 32 proteins with differential expression was visualized according to stimulation time using hierarchical clustering algorithms. Protein names (National Center for Biotechnology Information (NCBI)) are displayed for each cluster.

**Figure 4 toxics-11-00859-f004:**
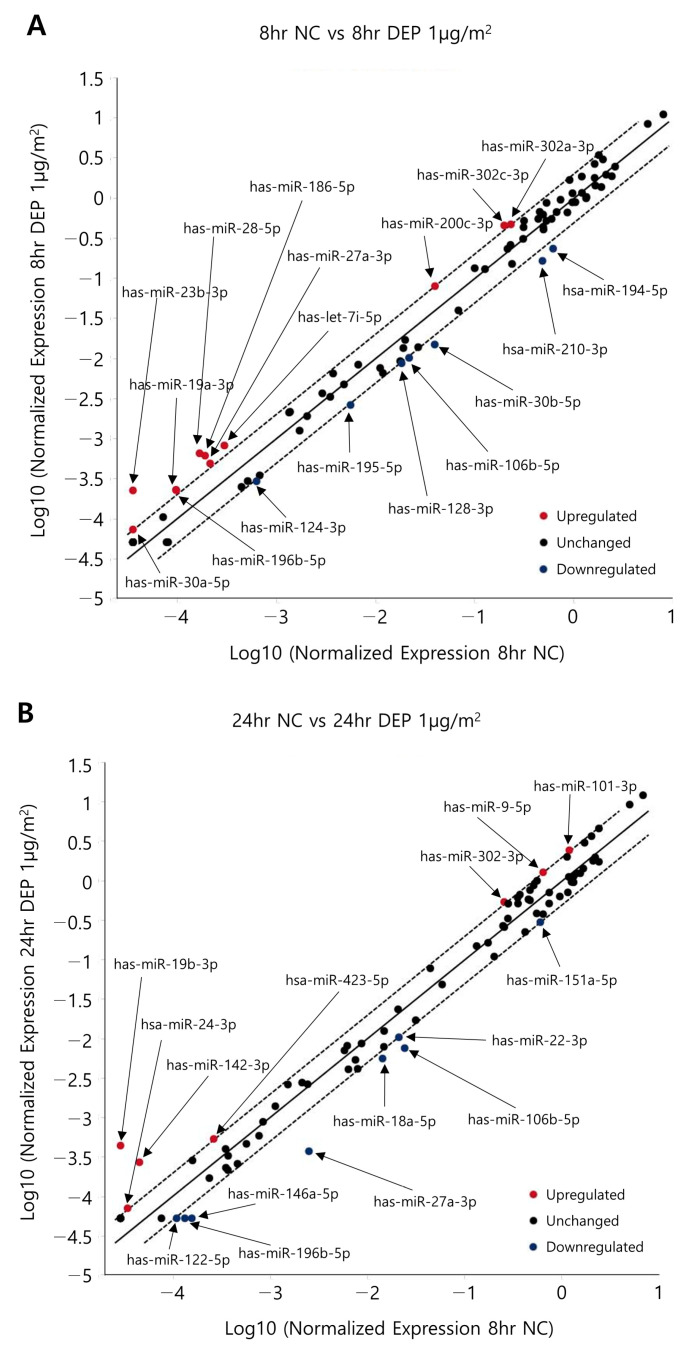
Scatter plots showing the differential expression of miRNAs in DEP-exposed NHBE cells. Scatter plots showing the miRNA expression levels of NHBE cells exposed to DEPs for (**A**) 8 and (**B**) 24 h.

**Figure 5 toxics-11-00859-f005:**
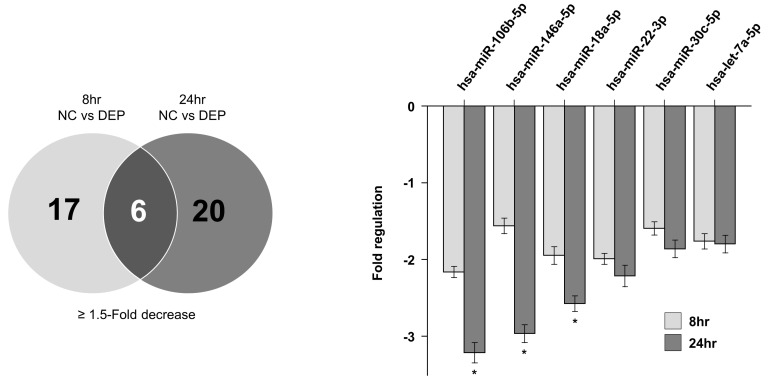
Overlap of miRNA in DEP-exposed NHBE cells. hsa-miR-146a-5p, hsa-miR18a-5p, hsa-miR,22-3p, hsa-miR-30c-5p, and hsa-let-7a-5p levels were decreased in NHBE cells exposed to DEPs. * *p* < 0.05 vs. control group.

**Figure 6 toxics-11-00859-f006:**
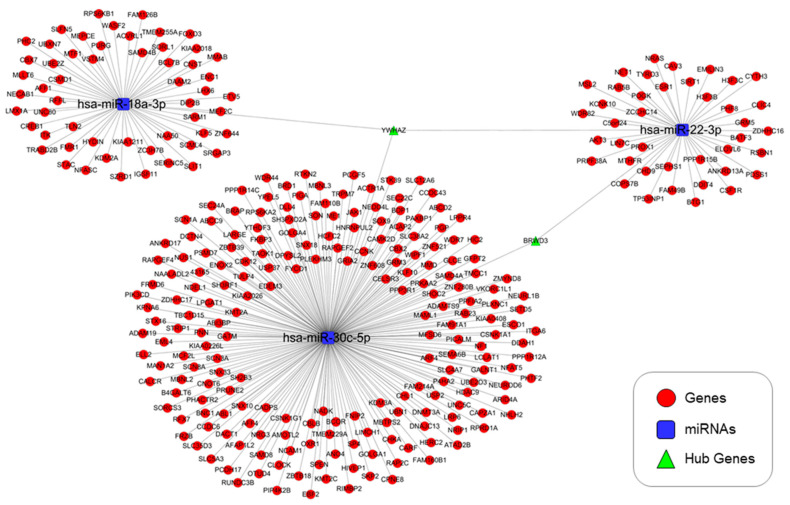
miRNA–mRNA networks of predicted regulation in DEP-exposed NHBE cells. Network visualization for miRNA–mRNA study was performed using Cytoscape software (version: 3.5.1, https://cytoscape.org/ (accessed on 25 September 2020)).

**Figure 7 toxics-11-00859-f007:**
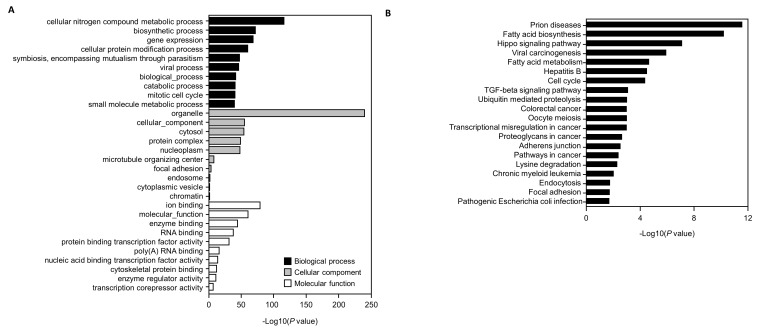
Gene ontology (GO) analysis of potential predicted genes and Kyoto Encyclopedia of Genes and Genomes database (KEGG) pathway enrichment analysis performed using DIANA-miRPath. (**A**) The 10 most significant genes for each GO enrichment term, including molecular functions, biological processes, and cellular components. (**B**) KEGG pathway analysis associated with predicted genes of the top 20 highly enriched pathways.

**Figure 8 toxics-11-00859-f008:**
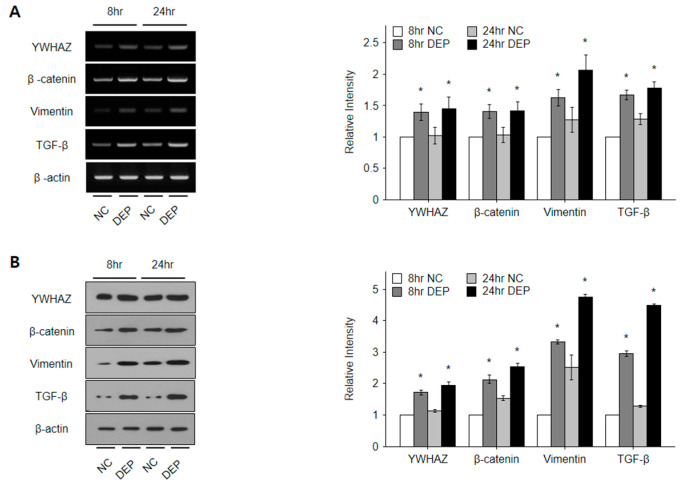
*YWHAZ*, *β-catenin*, *vimentin*, and *TGF-β* gene mRNA and proteins levels in epithelial cells. Band intensity in densitometry analysis graphs for (**A**) mRNA and (**B**) protein expression of *YWHAZ*, *β-catenin*, *vimentin*, and *TGF-β* was normalized to *β-actin*. * *p* < 0.05 vs. the NC group.

**Figure 9 toxics-11-00859-f009:**
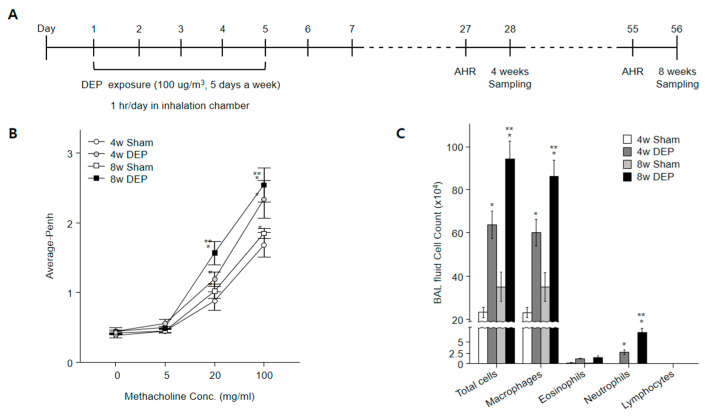
DEP exposure in mice increased airway inflammation, AHR, and differential cell count in BALF. (**A**) Experimental protocol for DEP exposure in mice (*n* = 30 in each group). (**B**) DEP nebulizer treatment increased AHR in mice. Penh was measured following increasing doses of methacholine. (**C**) Numbers of total cells, macrophages, eosinophils, neutrophils, and lymphocytes in BALF. * *p* < 0.05 compared with sham group. ** *p* < 0.05 compared with 4w DEP group.

**Figure 10 toxics-11-00859-f010:**
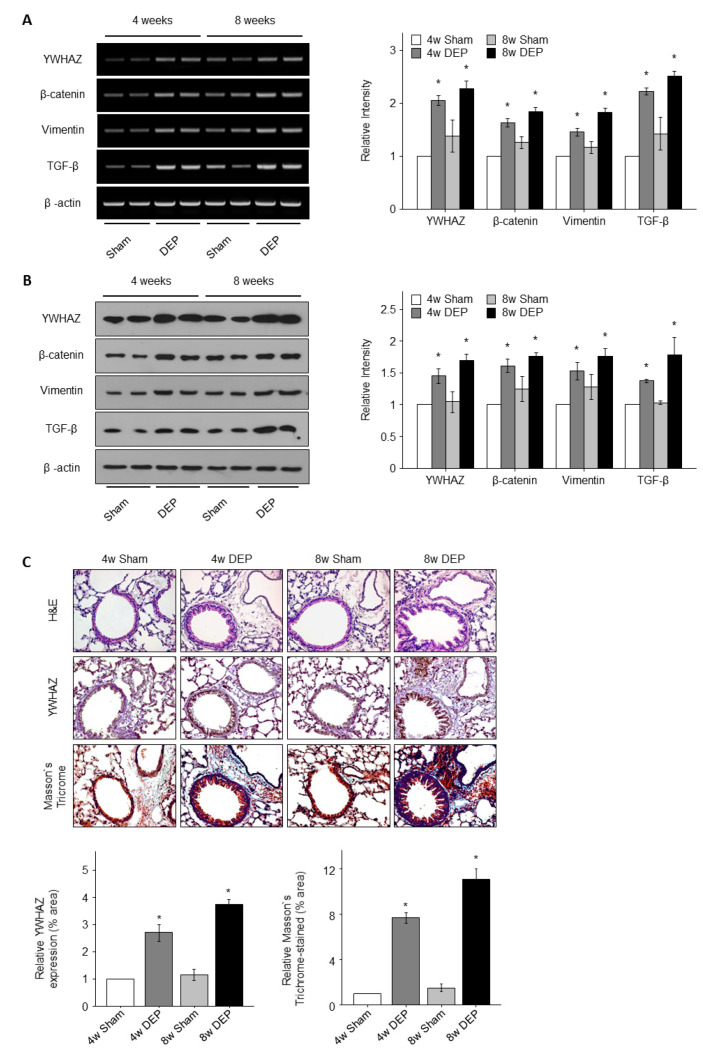
*YWHAZ*, *β-catenin*, *vimentin*, and *TGF-β* gene mRNA and proteins levels in mouse lung tissue. Band intensity in densitometry analysis graphs for (**A**) mRNA and (**B**) protein expression of *YWHAZ*, *β-catenin*, *vimentin*, and *TGF-β* was normalized to *β-actin* in mouse lung tissue. (**C**) Immunohistochemistry (IHC) stain and Masson trichrome stain of mouse lung tissue using antibodies for *YWHAZ*. Quantitation of the *YWHAZ* expression area intensity. Fibrosis as shown by Masson trichrome stain. * *p* < 0.05 vs. the sham group.

**Figure 11 toxics-11-00859-f011:**
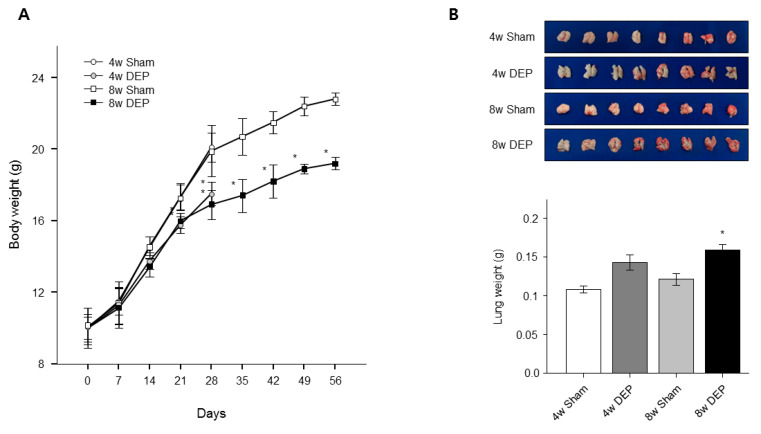
DEP-exposed mice were sacrificed after 4 weeks and 8 weeks, and the lung weight and body weight of each mouse were measured. Mouse model of DEP inhalation. (**A**) Body weight loss and (**B**) increased lung weight. * *p* < 0.05 compared with sham group.

**Figure 12 toxics-11-00859-f012:**
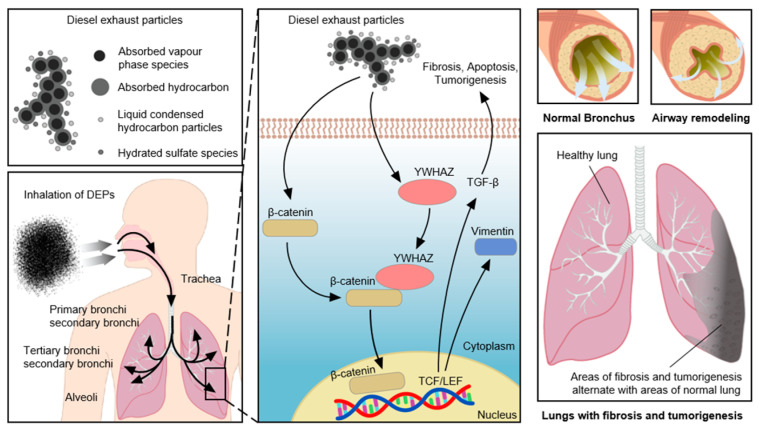
Summary of *YWHAZ*, *β-catenin*, *vimentin*, and *TGF-β* signaling and airway inflammation induced by DEP exposure in the lung.

**Table 1 toxics-11-00859-t001:** Primer design for RT-PCR.

Gene	Primer Sequence (5′→3′)	
Human		
YWHAZ	F: TGATCCCCAATGCTTCACAAG	R: GCCAAGTAACGGTAGTAATCTCC
β-catenin	F: GTGCTATCTGTCTGCTCTAGTA	R: CTTCCTGTTTAGTTGCAGCATC
Vimentin	F: AGGAAATGGCTCGTCACCTTCGTGAATA	R: GGAGTGTCGGTTGTTAAGAACTAGAGCT
TGF-β	F: TCCTGCTTCTCATGGCCA	R: CCTCAGCTGCACTTGTAG
β-Actin	F: TGCTGTCCCTGTATGCCTCT	R: CTTTGATGTCACGCACGATTT
Mouse		
YWHAZ	F: GAAAAGTTCTTGATCCCCAATGC	R: TGTGACTGGTCCACAATTCCTT
β-catenin	F: ATGGAGCCGGACAGAAAAGC	R: CTTGCCACTCAGGGAAGGA
Vimentin	F: CCCTCACCTGTGAAGTGGAT	R: TCCAGCAGCTTCCTGTAGGT
TGF-β	F: CACCGGAGAGCCCTGGATA	R: TGTACAGCTGCCGCACACA
β-Actin	F: CGGTTCCGATGCCCTGAGGCTCTT	R: CGTCACACTTCATGATGGAATTGA

**Table 2 toxics-11-00859-t002:** List of proteins identified via LC-MS/MS analysis—8 h NC vs. 8 h DEPs (1 ug/cm^2^) and 24 h NC vs. 24 h DEPs (1 ug/cm^2^).

No.	Protein Name	Symbol	Functional Categorization	Accession Number (NCBI)	Amino Acid Sequence	pI/Molecular Mass (Da)	Sequence Coverage (%)	Ratio	
								8 h	24 h
1	Prolyl 4-hydroxylase subunit beta	P4HB	Multifunctional enzyme	190384	K.SNFAEALAAHK.Y	4.76/57,487	28	+1.94	+2.14
2	Calreticulin	CALR	Calcium-binding protein	117501	K.GLQTSQDAR.F	4.29/48,286	2	+1.87	+1.98
3	Moesin	MSN	Cytoskeleton	127234	K.VTAQDVR.K	6.08/67,894	13	+2.37	+2.46
4	TGF-Beta Activated Kinase 1 (MAP3K7) Binding Protein 2	TAB2	TGF-beta signaling	7677466	U R.KNQIEIK.L	8.80/77,027	1	+1.86	+2.62
5	Karyopherin (importin) beta 1	KPNB1	Energy-dependent process	893288	R.VLANPGNSQVAR.V	4.68/98,442	1	+1.75	+2.79
6	Glucose-regulated protein	-	Glucose regulation	6900104	R.VEIIANDQGNR.I	5.07/72,404	30	+1.19	+2.13
7	Stratifin (14-3-3 sigma)	SFN (YWHAS)	Regulator of mitotic translation	398953	K.SNEEGSEEKGPEVR.E	4.68/27,873	44	+2.28	+3.02
8	Tyrosine 3-monooxygenase/tryptophan 5-monooxygenase activation protein, beta (14-3-3 beta)	YWHAB	Signal transduction	1345590	K.LAEQAER.Y	4.76/28,181	15	+1.32	+2.06
9	Tyrosine 3-monooxygenase/tryptophan 5-monooxygenase activation protein zeta (14-3-3 zeta)	YWHAZ	Signal transduction	30354619	K.SVTEQGAELSNEER.N	6.97/35,546	4	+2.24	+3.03
10	Tropomyosin 3	TPM3	Actin-binding protein	12653955	R.KIQVLQQQADDAEER.A	4.75/29,243	10	+1.76	+2.33
11	Tumor necrosis factor type 1 receptor associated protein	TRAP1	Tumor necrosis	687237	R.GVVDSEDIPLNLSR.E	8.05/80,251	1	+3.33	+4.14
12	60 kDa heat shock protein	HSP60	Macromolecular assembly	129379	U R.TVIIEQSWGSPK.V	5.70/61,190	21	+2.57	+3.00
13	Alpha tubulin	TUB1	Cytoskeleton	37492	U K.DVNAAIATIK.T	5.02/50,822	2	+2.10	+2.34
14	T cell receptor alpha	TRA	T-lymphocyte signaling	902377536	K.GITLSVRP.-	9.45/7206	12	+2.34	+2.79
15	Vimentin	VIM	Cytoskeleton	340219	R.SLYASSPGGVYATR.S	5.03/53,739	9	+2.38	+3.23
16	Phospholipase A2	PLA2	Phosphorylation	189953	K.SVTEQGAELSNEER.N	4.73/27,902	5	+2.24	+2.16
17	Annexin A2	ANXA2	Phospholipid-binding protein	113950	R.DALNIETAIK.T	7.57/38,808	25	+1.31	+1.35
18	S100 Calcium Binding Protein A9	S100A9	Calcium-binding protein	115444	K.LGHPDTLNQGEFKELVR.K	5.71/13,291	21	+3.08	+2.42
19	Cytovillin 2	VIL2	Cytoskeleton	340217	K.IALLEEAR.R	5.80/68,235	5	+2.58	+2.67
20	Heat shock cognate 71 kDa protein	HSP71	Macromolecular assembly	32467	R.TTPSYVAFTDTER.L	5.37/71,086	2	+1.20	+1.15
21	Enolase 2	ENO2	Glycolytic enzyme	119339	R.IGAEVYHNLK.N	5.78/49,852	46	+1.62	+1.54
22	Eukaryotic initiation factor 4A-I	EIF4A1	Eukaryotic initiation	219403	K.GYDVIAQAQSGTGK.T	5.32/46,357	8	+1.25	+1.15
23	Tropomyosin 3	TPM3	Actin-binding protein	12653955	R.KLVIIEGDLER.T	4.75/29,243	7	+1.78	+2.52
24	Calgizzarin (S100 calcium binding protein A11)	S100A11	Calcium-binding protein	560791	K.NQKDPGVLDR.M	6.56/11,849	9	+1.79	+1.49
25	Annexin A3	ANXA3	Phospholipid-binding protein	113954	R.DYPDFSPSVDAEAIQK.A	5.63/36,527	22	+1.87	+2.23
26	Phosphoglycerate mutase 1	PGAM1	Bisphosphoglycerate mutase activity	130348	R.HGESAWNLENR.F	6.67/28,900	15	+1.93	+1.88
27	Dermcidin	DCD	Proteolysis induction	20141302	K.ENAGEDPGLAR.Q	6.08/11,391	10	+1.84	+1.51
28	28 kDa heat shock protein	HSP28	Macromolecular assembly	433598	U R.QLSSGVSEIR.H	5.98/22,826	18	+1.65	+1.51
29	Keratin 9	KRT 9	Cytoskeleton	435476	R.SGGGGGGGLGSGGSIR.S	5.19/62,320	2	+1.85	+2.14
30	Keratin 19	KRT19	Cytoskeleton	6729681	R.QSSATSSFGGLGGGSVR.F	9.30/12,193	37	−1.15	−1.13
31	Dynactin 2	DCTN2	Cell division	12653855	K.YADLPGIAR.N	5.10/44,320	11	−1.13	−1.16
32	Chloride intracellular channel protein 1	CLIC1	Chloride ion channels	12643390	K.GVTFNVTTVDTK.R	5.09/27,254	23	−1.62	−1.63

**Table 3 toxics-11-00859-t003:** Summary of changes in miRNA levels in DEP-exposed NHBE cells.

microRNA	8 h		24 h	
	Fold Change	*p* Value	Fold Change	*p* Value
hsa-miR-142-5p	1.54	0.000001	2.13	0.000002
hsa-miR-9-5p	1.31	0.000012	2.06	0.000003
hsa-miR-150-5p	1.69	0.000019	1.48	0.000070
hsa-miR-27b-3p	1.41	0.000026	1.85	0.000047
hsa-miR-101-3p	1.91	0.000006	2.11	0.000026
hsa-let-7d-5p	1.98	0.000003	1.96	0.000017
hsa-miR-103a-3p	1.58	0.000002	1.82	0.000002
hsa-miR-16-5p	1.29	0.000019	1.66	0.000010
hsa-miR-26a-5p	−2.46	0.000002	−2.1	0.000001
hsa-miR-32-5p	−1.31	0.000001	−1.04	0.001032
hsa-miR-26b-5p	1.57	0.000001	1.87	0.000003
hsa-let-7g-5p	1.19	0.000005	−1.05	0.001569
hsa-miR-30c-5p	−1.98	0.000001	−1.89	0.000000
hsa-miR-96-5p	1.09	0.001910	1.09	0.006373
hsa-miR-185-5p	1.36	0.000143	1.13	0.012914
hsa-miR-142-3p	−1.91	0.000880	6.5	0.000334
hsa-miR-24-3p	1.44	0.015895	2.15	0.008468
hsa-miR-155-5p	−1.07	0.164969	1.1	0.140376
hsa-miR-146a-5p	−1.76	0.000003	−2.01	0.000001
hsa-miR-425-5p	1.68	0.000012	2.03	0.000009
hsa-miR-181b-5p	−1.16	0.000013	−1.15	0.000344
hsa-miR-302b-3p	−1.31	0.000006	−1.31	0.000009
hsa-miR-30b-5p	1.27	0.000023	1.21	0.000108
hsa-miR-21-5p	1.51	0.000014	1.85	0.000047
hsa-miR-30e-5p	1	0.605624	1.2	0.000037
hsa-miR-200c-3p	2.07	0.000086	1.8	0.000183
hsa-miR-15b-5p	1.14	0.000707	1.01	0.463162
hsa-miR-223-3p	−1.04	0.000557	−1.17	0.000000
hsa-miR-194-5p	−2.74	0.000000	−4.58	0.000000
hsa-miR-210-3p	−3.06	0.000000	−3.94	0.000000
hsa-miR-15a-5p	1.6	0.001296	1.32	0.013510
hsa-miR-181a-5p	−1.08	0.000326	−1.17	0.000032
hsa-miR-125b-5p	−1.17	0.000001	−1.65	0.000000
hsa-miR-99a-5p	1.41	0.000004	1.82	0.000000
hsa-miR-28-5p	4.12	0.000502	−1.23	0.100464
hsa-miR-320a	1.54	0.000001	1.91	0.000000
hsa-miR-125a-5p	1.68	0.000002	1.82	0.000006
hsa-miR-29b-3p	−1.3	0.000027	−1.11	0.000585
hsa-miR-29a-3p	1.02	0.176756	−1.06	0.056638
hsa-miR-141-3p	−1.39	0.000006	−1.17	0.000139
hsa-miR-19a-3p	2.44	0.000933	1.16	0.235469
hsa-miR-18a-5p	−2.43	0.000007	−2.83	0.000007
hsa-miR-374a-5p	−1.93	0.000017	−2.83	0.000010
hsa-miR-423-5p	1.47	0.007195	2.12	0.006269
hsa-let-7a-5p	−1.78	0.000324	−1.82	0.000207
hsa-miR-124-3p	−2.22	0.000090	−1.32	0.007744
hsa-miR-92a-3p	−1.35	0.000001	−1.4	0.000003
hsa-miR-23a-3p	−1.23	0.000000	−1.73	0.000001
hsa-miR-25-3p	1.1	0.000004	−1.25	0.000019
hsa-let-7e-5p	−1	0.747350	−1.45	0.000000
hsa-miR-376c-3p	−1.09	0.000008	−1.53	0.000001
hsa-miR-126-3p	−2.22	0.000007	−6.87	0.000002
hsa-miR-144-3p	−1.09	0.000005	1.79	0.000000
hsa-miR-424-5p	−1.18	0.002114	1.15	0.013081
hsa-miR-30a-5p	2.72	0.002933	−1.87	0.009875
hsa-miR-23b-3p	8.6	0.001428	−1.13	0.516472
hsa-miR-151a-5p	−1.02	0.010708	−2.06	0.000000
hsa-miR-195-5p	−2.21	0.000022	−1	0.952589
hsa-miR-143-3p	−1.06	0.205299	−1.96	0.000574
hsa-miR-30d-5p	−1.06	0.001098	−1.4	0.000020
hsa-miR-191-5p	−1.02	0.642426	1.23	0.012369
hsa-let-7i-5p	2.86	0.000598	1.05	0.595219
hsa-miR-302a-3p	2.09	0.000032	1.88	0.000062
hsa-miR-222-3p	−1.38	0.001276	1.77	0.000526
hsa-let-7b-5p	−1.84	0.000341	−4.86	0.000134
hsa-miR-19b-3p	1.44	0.015895	21.69	0.001596
hsa-miR-17-5p	1.44	0.015895	1.89	0.002513
hsa-miR-93-5p	1.44	0.015895	1.89	0.002513
hsa-miR-186-5p	3.36	0.000553	1.85	0.001160
hsa-miR-196b-5p	2.4	0.000872	−3	0.000250
hsa-miR-27a-3p	2.36	0.000702	−7.22	0.000955
hsa-miR-22-3p	−2.08	0.000008	−2.03	0.000005
hsa-miR-130a-3p	−1.84	0.001210	−3.62	0.000328
hsa-let-7c-5p	−3.18	0.000116	−3.15	0.000490
hsa-miR-29c-3p	1.6	0.001198	1.25	0.006436
hsa-miR-140-3p	1.28	0.006225	−1.59	0.002090
hsa-miR-128-3p	−2.17	0.000007	−2.84	0.000001
hsa-let-7f-5p	1.44	0.015895	1.89	0.002513
hsa-miR-122-5p	−1.49	0.000107	−1.17	0.002218
hsa-miR-20a-5p	1.26	0.003875	−1.42	0.000638
hsa-miR-106b-5p	−2.05	0.000160	−1.74	0.000260
hsa-miR-7-5p	1.78	0.000549	1.33	0.010385
hsa-miR-100-5p	−1.13	0.000001	1.3	0.000009
hsa-miR-302c-3p	2.38	0.000029	2.19	0.000075

**Table 4 toxics-11-00859-t004:** Summary of changes in miRNA levels in DEP-exposed NHBE cells.

Sample	miRNAs Detected	≥1.5-Fold Increase (*n*)	≥1.5-Fold Decrease (*n*)	≥2-Fold Increase (*n*)	≥2-Fold Decrease (*n*)
8 h NC vs. 8 h DEPs	84	22	17	11	7
24 h NC vs. 24 h DEPs		26	20	7	8
Overlapping miRNAs		6	6	1	4

## Data Availability

The authors confirm that all data underlying the findings are fully available without restriction. All relevant data are within the paper.

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
