# Peer review of "Analyzing the Impact of Diesel Exhaust Particles on Lung Fibrosis Using Dual PCR Array and Proteomics: YWHAZ Signaling"

_toxics, 2023, doi:10.3390/toxics11100859_

Round 1

Reviewer 1 Report

The authors applied a number of technologies such as PCR and LC-MS/MS proteomics analysis to investigate the impact of DEP on respiratory diseases by looking at both RNA and protein levels. I have some questions mostly in proteomics sections.

In Table 1, it lists the protein ID and sequence coverage from the proteomics analysis. However, the coverage is quite low, less than 50%. Normally, protein samples from cells or tissues need to be extracted and go through trypsin or chymotrypsin digestion for peptides identification. I do not see the digestion experiment being performed in the study. This may explain a very low coverage of sequence and may affect the accuracy of the sequence coverage. If the authors performed trypsin digestion, a detailed experimental protocol needs to be provided in the manuscript.

During LC-MS/MS analysis, please mention how much protein sample in nanogram was injected per run. What is the injection volume in micro liter?

English is sufficient.

Author Response

Manuscript ID: Toxics-2633547

Manuscript Title: The Impact of Diesel Exhaust Particles on Lung Fibrosis Using Dual PCR Array and Proteomics: YWHAZ Signaling

Dear Editor:

We wish to express our appreciation the opportunity to revise and re-submit our manuscript. We enclose a point by point response to the reviewers’ comments. We believe the revised manuscript now fulfills the high standards of Toxics journal.

With my best regards,

An-Soo Jang, M.D., Ph.D.

Department of Internal Medicine, Soonchunhyang University Bucheon Hospital,170 Jomaru-ro, Wonmi-gu, Bucheon, Gyeonggi-Do, 14584, South Korea

Telephone: +82-32-621-5143, FAX: +82-32-621-6950, E-mail: jas877@schmc.ac.kr

Reviewer 1 Review Report (Round 1)

The authors applied a number of technologies such as PCR and LC-MS/MS proteomics analysis to investigate the impact of DEP on respiratory diseases by looking at both RNA and protein levels. I have some questions mostly in proteomics sections.

C1. In Table 1, it lists the protein ID and sequence coverage from the proteomics analysis. However, the coverage is quite low, less than 50%. Normally, protein samples from cells or tissues need to be extracted and go through trypsin or chymotrypsin digestion for peptides identification. I do not see the digestion experiment being performed in the study. This may explain a very low coverage of sequence and may affect the accuracy of the sequence coverage. If the authors performed trypsin digestion, a detailed experimental protocol needs to be provided in the manuscript.

R1. Thank you for your point. The authors have changed text in methods section as you suggest.

  • Peptides obtained after trypsin digestion from NHBE cell lysates were analyzed with a Nano LC-MS/MS.

C2. During LC-MS/MS analysis, please mention how much protein sample in nanogram was injected per run. What is the injection volume in micro liter?

R2. Thank you for your point. The authors have changed text in methods section as you suggest.

  • Samples digest were dissolved in 0.1% TFA and 1ul (equivalent to 2 ug) was injected on a C18 nanobore column (150 mm × 0.1 mm, 3 μm pore size; Agilent).

Thank you

Reviewer 2 Report

The authors report data from PCR arrays and proteomics analysis of NHBE cells and BalbC mice lungs exposed to diesel exhaust particles. The data are consistent with the findings of  "complementary PCR and proteomic approaches identified signaling pathways 447 triggering YWHAZ, β-catenin, vimentin, and TGF-β overexpression in lung tissue."  The manuscript is consistent with other prior manuscripts showing diesel exhaust particles stimulate toxic responses in human airway epithelium in vitro and in vivo in mouse lungs.  The attempt to add value by doing a dual miRNA-mRNA PCR array approach vs proteomics adds novelty and is a good strategy. However, there are several points that detract from the publication in its current form.

Major concerns:

1. Extensive English grammatical errors occur throughout the manuscript, mostly with verb choice or with simply incomplete sentences often missing subject nouns.

2. The authors state that at least two experiments with internal duplicates were performed for each assay. Proper statistics usually requires at least an n>3 for the NHBE experiments. For PCR and LC-MS/MS analyses,  have any correction factors been used in the statistical analysis to account for multiple endpoints from the same samples, since n =2 experiments already creates a probelm? In contrast, the rodent experiments are more than adequately powered at n=30. 

3. Using NHBE cells, it is always a trade off between using the BulletKit supplements to sustain robust cell growth verses converting the cells to a basal media 24-48 hours before stimulation.  It may be difficult to see agonist-induced changes in the presence of over a dozen other biologics that are part of the supplements. This may underestimate the changes induced by DEP. 

4. Given that the mouse experiments tested many weeks of exposure to DEPs ( a good model for air pollutants), why were 8 and 24 hours the only endpoints for NHBE cells in tissue culture? Also, can you comment on how you reconcile changes in miRNA, mRNA and protein over different time points given the different half-lives of miRNA, mRNA and protein, in either the in vitro or in vivo analyses. The non-parametric tests applied are normally fine for control vs agonist, but digging deeper into causal mechanisms in pathways analysis (like the GO analysis) can be tricky with just 8 vs 24 hour timepoints. Care must be taken to leap from NHBE to mouse lungs. 

Minor concerns.

1. The choice of 2D - LC-MS/MS is interesting. Was untargeted LC-MS/MS available for cell extracts? It would likely show more quantifiable DEP induced changes than afforded by 2D IEF-PAGE. 

2. Figure 4 and Figure 6 are hard to read due to faint and small font choices. Many analysis software tools simply do not make great figures for publication.  Can you do better? Otherwise, what is the information you are trying to relay in each figure? Would a volcano plot or PCA help for figure 4 where the 1.5 and 2 fold changes are hard to interpret? Would other network analyses better emphasize your hub genes and miRNAs? The many unreadable individual genes dominate the visual effect of that figure even though they cannot be read easily.   

Many sentences are missing words, they are not acceptable in English language. Subject nouns or object nouns are often missing, such that the meaning of sentence is sometimes vague. Verb tenses in English can be a challenge also. Please have another good English language edit performed.

Author Response

Manuscript ID: Toxics-2633547

Manuscript Title: The Impact of Diesel Exhaust Particles on Lung Fibrosis Using Dual PCR Array and Proteomics: YWHAZ Signaling

Dear Editor:

We wish to express our appreciation the opportunity to revise and re-submit our manuscript. We enclose a point by point response to the reviewers’ comments. We believe the revised manuscript now fulfills the high standards of Toxics journal.

With my best regards,

An-Soo Jang, M.D., Ph.D.

Department of Internal Medicine, Soonchunhyang University Bucheon Hospital,170 Jomaru-ro, Wonmi-gu, Bucheon, Gyeonggi-Do, 14584, South Korea

Telephone: +82-32-621-5143, FAX: +82-32-621-6950, E-mail: jas877@schmc.ac.kr

Reviewer 2 Review Report (Round 1)

The authors report data from PCR arrays and proteomics analysis of NHBE cells and BalbC mice lungs exposed to diesel exhaust particles. The data are consistent with the findings of  "complementary PCR and proteomic approaches identified signaling pathways 447 triggering YWHAZ, β-catenin, vimentin, and TGF-β overexpression in lung tissue."  The manuscript is consistent with other prior manuscripts showing diesel exhaust particles stimulate toxic responses in human airway epithelium in vitro and in vivo in mouse lungs.  The attempt to add value by doing a dual miRNA-mRNA PCR array approach vs proteomics adds novelty and is a good strategy. However, there are several points that detract from the publication in its current form.

Major concerns:

C1. Extensive English grammatical errors occur throughout the manuscript, mostly with verb choice or with simply incomplete sentences often missing subject nouns.

R1. We checked English by English editing servece.

From : service@textcheck.com To : byeonggone@naver.com

Subject : Textcheck - 18122608: Completed

Dear Dr Kim,

Re: PCR array Dear Dr Kim,

Re: PCR array and proteomic dual approach to identify genes and proteins expressed in human bronchial epithelial cells exposed to diesel exhaust particles

Please find attached your completed document. The same files are also available in your account on our www site.

Two files are attached. Please note that 18122608f.doc is your finished

document (see notes below re 18122608r.doc).

Please read the notes provided at the end of 18122608f.doc. 

For more information please see 'When you receive your completed document' at http://www.textcheck.com/text/page/guidelines

If there are any points that the editor has misunderstood, just rewrite

the sentence(s) in 18122608f.doc and upload the file to our WWW site. You do not need to mark the changes.   (DO NOT USE 18122608r.doc - see below).

If you have no questions, then please use our WWW site to obtain an invoice by download, e-mail, or airmail.

To login, go to: http://www.textcheck.com/login

Click on 'Billing'. Thank you for using Textcheck. Yours sincerely, Richard Turner

C2. The authors state that at least two experiments with internal duplicates were performed for each assay. Proper statistics usually requires at least an n>3 for the NHBE experiments. For PCR and LC-MS/MS analyses, have any correction factors been used in the statistical analysis to account for multiple endpoints from the same samples, since n =2 experiments already creates a probelm? In contrast, the rodent experiments are more than adequately powered at n=30.

R2. The authors agree with you. In cell experiments, various experiments such as cell viability, PCR, western blot, etc. were performed with a minimum of n>3 replicates. The analysis of proteomics involved the use of statistical analysis on multiple endpoints from the same samples. To obtain reliability of analyzed data such as lung capacity, BALF, histology, and tissue, data were extracted and analyzed through replicates. The number of mice inhaled by the size of the chamber is limited. We have conducted repeated experiments to generate additional experimental data by measuring and analyzing lung capacity and body weight, etc. Each mouse was experimented with a relatively large number of mice due to the high standard deviation caused by various factors.

C3. Using NHBE cells, it is always a trade off between using the BulletKit supplements to sustain robust cell growth verses converting the cells to a basal media 24-48 hours before stimulation.  It may be difficult to see agonist-induced changes in the presence of over a dozen other biologics that are part of the supplements. This may underestimate the changes induced by DEP.

R3. Thank you for your comments. This can be interpreted as a protein that exhibits a significant variation compared to the proteins that have been underestimated. It was conducted to assess the impact of DEP while maintaining cell growth conditions by considering data from previous studies.

(References)

Kim, B.G., et al. Impact of ozone on claudins and tight junctions in the lungs. Environ Toxicol. 2018. 33:798–806.

Kim, B.G., et al. Effects of nanoparticles on neuroinflammation in a mouse model of asthma. Respir Physiol Neurobiol. 2020. 271:103292.

C4. Given that the mouse experiments tested many weeks of exposure to DEPs ( a good model for air pollutants), why were 8 and 24 hours the only endpoints for NHBE cells in tissue culture? Also, can you comment on how you reconcile changes in miRNA, mRNA and protein over different time points given the different half-lives of miRNA, mRNA and protein, in either the in vitro or in vivo analyses. The non-parametric tests applied are normally fine for control vs agonist, but digging deeper into causal mechanisms in pathways analysis (like the GO analysis) can be tricky with just 8 vs 24 hour timepoints. Care must be taken to leap from NHBE to mouse lungs.

R4. Thank you for your coments. Based on previous research data, the time difference in inflammation was set before cells rapidly died due to exposure to particles. This is because if the time difference is small, the toxic effect caused by particles does not change significantly. In the case of animal models, there was an effect on recovery and resistance.

(Reference)

Kim, B.G., et al. Effects of nanoparticles on neuroinflammation in a mouse model of asthma. Respir Physiol Neurobiol. 2020. 271:103292.

Kim, B.G., et al. Long-Term Effects of Diesel Exhaust Particles on Airway Inflammation and Remodeling in a Mouse Model. Allergy Asthma Immunol Res. 2016. 8:246–256.

Minor concerns.

C1. The choice of 2D - LC-MS/MS is interesting. Was untargeted LC-MS/MS available for cell extracts? It would likely show more quantifiable DEP induced changes than afforded by 2D IEF-PAGE.

R1. Thank you for your point. We selected and analyzed the protein induced by DEP exposure based on 2DE when designing protein analysis, so we are going to consider it in further study.

C2. Figure 4 and Figure 6 are hard to read due to faint and small font choices. Many analysis software tools simply do not make great figures for publication.  Can you do better? Otherwise, what is the information you are trying to relay in each figure? Would a volcano plot or PCA help for figure 4 where the 1.5 and 2fold changes are hard to interpret? Would other network analyses better emphasize your hub genes and miRNAs? The many unreadable individual genes dominate the visual effect of that figure even though they cannot be read easily.

R2. Thank you for your point. The authors have changed the picture to be easy to read as the you suggest.

Thank you

Round 2

Reviewer 2 Report

1. The figures are much improved by simply enlarging them to allow the readers to see the fonts sufficiently to understand what the authors intended to communicate.

2. While the authors offer some explanations of duplicates and statistics, some of their rebuttal would help clarify the manuscript if included. It is important to be as clear a possible how experimental and analytical decisions were made.  The manuscript is sufficient, but future manuscripts can be improved by educating the readers as to choices made in experimental designs. 

3. What does line 256 mean? The data were entered twice into SPSS?

There are still minor and major English issues, for example line 10 of the abstract should read "not enough...", "no enough" is not technically proper English.  The editing service you used did not do well. 

Line 14 "both miRNA and protein expressed identified." is awkward English. How about saying "both miRNA and protein expression was identified", or use a more active voice "Polymerase chain reaction (PCR) array and 2DE LC-MS/MS proteomics were used to identify changes in miRNA and protein expression, respectively." 

And line 14 "Gene that discovered through overlap analysis validated in mice model exposed to DEP" is not a good English sentence.  What gene are you talking about in this sentence? Please correct this type of mistake. 

There are still others in the manuscript, do not rely solely on the editing service. Perhaps find a good English editor in the University to assist you.

Author Response

Manuscript ID: Toxics-2633547

Manuscript Title: The Impact of Diesel Exhaust Particles on Lung Fibrosis Using Dual PCR Array and Proteomics: YWHAZ Signaling

Dear Editor:

We wish to express our appreciation the opportunity to revise and re-submit our manuscript. We enclose a point by point response to the reviewers’ comments. We believe the revised manuscript now fulfills the high standards of Toxics journal.

With my best regards,

An-Soo Jang, M.D., Ph.D.

Department of Internal Medicine, Soonchunhyang University Bucheon Hospital,170 Jomaru-ro, Wonmi-gu, Bucheon, Gyeonggi-Do, 14584, South Korea

Telephone: +82-32-621-5143, FAX: +82-32-621-6950, E-mail: jas877@schmc.ac.kr

Reviewer 2 Review Report (Round 2)

C1. The figures are much improved by simply enlarging them to allow the readers to see the fonts sufficiently to understand what the authors intended to communicate.

R1. Thank you for your comments. The authors have changed Figures.

C2. While the authors offer some explanations of duplicates and statistics, some of their rebuttal would help clarify the manuscript if included. It is important to be as clear a possible how experimental and analytical decisions were made.  The manuscript is sufficient, but future manuscripts can be improved by educating the readers as to choices made in experimental designs.

R2. The authors agree with you. In order to clarify the manuscript, we revised it by accepting our opinions and implementing further professional English corrections.

C3. What does line 256 mean? The data were entered twice into SPSS?

R3. Thank you for your point. We have changed followings in methods session. The authors conducted more professional English corrections.

Statistical analysis was performed using SPSS statistical software package (ver. 20.0; SPSS; Chicago, IL, USA).

Comments on the Quality of English Language

C4. There are still minor and major English issues, for example line 10 of the abstract should read "not enough...", "no enough" is not technically proper English.  The editing service you used did not do well.

R4. Thank you for your comments. The authors have changed abstract session.

C5. Line 14 "both miRNA and protein expressed identified." is awkward English. How about saying "both miRNA and protein expression was identified", or use a more active voice "Polymerase chain reaction (PCR) array and 2DE LC-MS/MS proteomics were used to identify changes in miRNA and protein expression, respectively."

R5. Thank you for your comments. The authors have changed abstract session.

C6. And line 14 "Gene that discovered through overlap analysis validated in mice model exposed to DEP" is not a good English sentence.  What gene are you talking about in this sentence? Please correct this type of mistake.

R6. Thank you for your comments. The authors have changed abstract session.

C7. There are still others in the manuscript, do not rely solely on the editing service. Perhaps find a good English editor in the University to assist you.

R7. Thank you for your comments. The authors agree with you. The authors conducted more professional English corrections.
